# Current and Future Applications of Artificial Intelligence in Coronary Artery Disease

**DOI:** 10.3390/healthcare10020232

**Published:** 2022-01-26

**Authors:** Nitesh Gautam, Prachi Saluja, Abdallah Malkawi, Mark G. Rabbat, Mouaz H. Al-Mallah, Gianluca Pontone, Yiye Zhang, Benjamin C. Lee, Subhi J. Al’Aref

**Affiliations:** 1Department of Internal Medicine, University of Arkansas for Medical Sciences, Little Rock, AR 72205, USA; ngautam@uams.edu (N.G.); psaluja@uams.edu (P.S.); 2Department of Medicine, Division of Cardiology, University of Arkansas for Medical Sciences, Little Rock, AR 72205, USA; amalkawi@uams.edu; 3Division of Cardiology, Loyola University Medical Center, Chicago, IL 60153, USA; mrabbat@lumc.edu; 4Houston Methodist DeBakey Heart & Vascular Center, Houston, TX 77030, USA; mouaz74@gmail.com; 5Centro Cardiologico Monzino IRCCS, 20138 Milan, Italy; gianluca.pontone@ccfm.it; 6Department of Population Health Sciences, Weill Cornell Medicine, New York, NY 10021, USA; yiz2014@med.cornell.edu; 7Department of Radiology, Dalio Institute of Cardiovascular Imaging, New York-Presbyterian Hospital and Weill Cornell Medicine, New York, NY 10021, USA; bcl2004@med.cornell.edu

**Keywords:** artificial intelligence, coronary artery disease, major adverse cardiovascular events, fractional flow reserve, cardiac computed tomography

## Abstract

Cardiovascular diseases (CVDs) carry significant morbidity and mortality and are associated with substantial economic burden on healthcare systems around the world. Coronary artery disease, as one disease entity under the CVDs umbrella, had a prevalence of 7.2% among adults in the United States and incurred a financial burden of 360 billion US dollars in the years 2016–2017. The introduction of artificial intelligence (AI) and machine learning over the last two decades has unlocked new dimensions in the field of cardiovascular medicine. From automatic interpretations of heart rhythm disorders via smartwatches, to assisting in complex decision-making, AI has quickly expanded its realms in medicine and has demonstrated itself as a promising tool in helping clinicians guide treatment decisions. Understanding complex genetic interactions and developing clinical risk prediction models, advanced cardiac imaging, and improving mortality outcomes are just a few areas where AI has been applied in the domain of coronary artery disease. Through this review, we sought to summarize the advances in AI relating to coronary artery disease, current limitations, and future perspectives.

## 1. Introduction

Clinically significant atherosclerosis of the coronary arteries, known as coronary artery disease (CAD), is an endemic condition that is associated with significant morbidity and mortality [1]. For instance, CAD is reported to have affected 20.1 million American adults between 2015 and 2018 [2]. Current societal guidelines emphasize the importance of early detection and risk stratification in the appropriate age and risk groups, with the goal of implementation of goal-directed medical therapies that can alter the natural trajectory of CAD to a less morbid course [3]. Traditional population-derived primary and secondary prevention cardiovascular risk assessment tools (e.g., Framingham risk score, ASCVD, TIMI score, GRACE score, etc.) have historically relied on patient-level data that are easily retrievable and practical to utilize in the clinical setting. Despite their importance, such tools are inherently limited by design due to relying on regression models that make many mathematical assumptions that often do not hold in a real-world setting, such as collinearity between variables and homogeneity of effects. The complex nature and multifactorial pathology of CAD make such regression-based tools less generalizable across different populations. 

Recently, the digitization of health records has improved access to large repositories of clinical and imaging datasets for clinical care and research purposes. This is coupled with advances in diagnostic tools that are available for the detection and quantification of CAD. To that end, recent studies have highlighted the usefulness of these tools in enhancing risk assessment and decision making through incorporation of different yet complementary findings from these imaging modalities (e.g., quantitative and qualitative plaque features on computed tomographic imaging of the coronary circulation, coupled with functional and physiologic findings on stress-test imaging). In addition, there has been an increasing interest in using the plethora of data in electronic health records and genomic data for better risk assessment [4]. Such tools are being integrated in practice as complementary methods to traditional tools [5,6,7].

Yet, despite the ever-increasing amounts of data, risk-prediction methods have been historically limited by what was possible with traditional statistical tools. The concept of Artificial Intelligence (AI) was introduced to mankind as early as the 1950s, with its employment in medical sciences commencing in the 1970s [8]. AI has gained momentum recently, fueled by an improvement in computational power, accumulation of data, and cloud processing. With the attempt to transfer a significant portion of human intelligence to machines, there has been a concerted effort aimed at harnessing the power of AI for biomedical applications in the past two decades [9,10]. Machine learning (ML) is a subfield of AI that involves the creation of algorithms that analyze large datasets without prior assumptions and learn rules and patterns between variables to make predictions and classifications [11]. On the other hand, deep learning (DL) is a subset of ML geared towards image analysis and utilizes more intricate algorithms known as neural networks with multiple deep, hidden layers. Specifically, while ML usually relies on structured data with handcrafted features often in tabular form, DL algorithms can input both structured and raw, unstructured data (e.g., images, video, and text) and extract their own features.

ML algorithms can incorporate a larger number of variables from different modalities, including both patient-level clinical parameters as well as two- and three-dimensional imaging data that take into account the multidimensional nonlinear interactions between variables [11]. Implementing such techniques in healthcare mainly aims to improve the accuracy of risk prediction and customize clinical decisions to each individual, which is the overarching theme in the goal of achieving precision medicine. In this paper, we summarize the recent advances in ML and current attempts at improving predictive analytics with relevance to CAD. We also elucidate on the role of AI in genetics, the incremental role of AI in improving post-procedure risk prediction and long-term mortality. Lastly, we discuss the limitations and potential near-future applications of AI within cardiovascular medicine.

## 2. Integration of Genetics and AI in Cardiovascular Diseases

Over the last two decades, the emergence of technologies able to measure biological processes at a large scale have resulted in an enormous influx of data. For instance, the completion of the Human Genome Project has paved the way to design single-nucleotide polymorphism (SNP) and mRNA microarrays, which can broadly test for millions of genetic variants in a simple point-of-care test. This has paved the way for the emergence of modern data-driven sciences such as genomics and other “omics” [12]. Genome-wide association studies (GWASs) operate by simultaneous comparison of millions of SNPs between diseased individuals and disease-free controls to detect a statistically significant association between an SNP locus and a particular condition [12]. Erdmann et al. reported that up until the year 2018, GWASs have successfully identified 163 distinct genetic loci for SNPs that are associated with CAD [13]. The risk for expressing a complex trait like CAD can be represented by a mathematical model that assumes a normal distribution of a binary outcome (i.e., CAD or no CAD) and captures the aggregate influence of multiple genetic variants that are predisposed to disease. Such a model is referred to as a polygenic risk score (PRS). PRSs were proposed early on to improve risk stratification in CAD risk models, especially when combined with traditional cardiovascular risk factors. However, the complex genetic architecture along with the multifactorial nature of CAD have been major challenges in CAD risk prediction [14]. For instance, Kathiresan et al. built a genetic risk score to predict major adverse cardiovascular events based on nine different dyslipidemia-related SNPs previously identified in GWASs. Adding the genetic score to a Cox proportional hazard model along with traditional risk factors did not improve predictive accuracy as measured by the *C* statistic model; however, there was a significant improvement in the net reclassification index, which accounts for correct movement of categories (assigning high-risk for patients who developed the disease, and low-risk for those who were disease-free) [15]. Brautbar et al. also suggested a genetic risk score to predict coronary heart disease based on SNPs. Adding the genetic risk score to traditional risk factors in a Cox proportional hazard model only modestly improved the area under the curve (AUC) for prediction of coronary heart disease from 0.742 to 0.749 (Δ = 0.007; 95% CI, 0.004–0.013) [16]. ML and particularly DL algorithms are inherently designed to extract patterns and associations from large-scale data, including clinical and genomic data. Given the complexity and multifaceted nature of cardiovascular diseases in general, and CAD in particular, an approach that integrates all these factors into a risk-stratification model would be expected to better predict incident events than existent models [17].

Multiple studies have emphasized the role of ML in identifying genetic variants and expression patterns associated with CAD from mRNA arrays using differential expression analysis and protein–protein interaction networks [18,19]. For example, Zhang et al. used ML to perform differential expression analysis on mRNA profiles from CAD patients and healthy controls to identify a set of differentially expressed genes between the two groups, then built a network representation of functional protein–protein interaction. The top 20 genes in the network were identified using a maximal clique centrality (MCC) algorithm. Finally, to test the performance, a logistic regression model was built using the top five predictor genes to classify individuals into the presence or absence of CAD. The model achieved an AUC of 0.9295 and 0.8674 in the training and internal validation sets respectively [20]. 

Dogan et al. built an ensemble model of eight random-forest (RF) classifiers to predict the risk of symptomatic CAD using genetic and epigenetic variables along with clinical risk factors. The model was trained on a cohort derived from the Framingham heart study (*n* = 1545) and utilized variables derived from genome-wide array chips to extract epigenetic (DNA methylation loci) and genetic (SNP) profiles. The initial number of available variables were 876,014 SNP and DNA methylation (CpG) loci, which required multiple reduction steps, ending up with 4 CpG and 2 SNP predictors fed into the model along with age and gender. The model predicted symptomatic CAD with an accuracy, sensitivity, and specificity of 0.78, 0.75, and 0.80, respectively, in the internal validation cohort (*n* = 142). For comparison, a similar ensemble model was built using clinical risk factors only as predictor variables and had an accuracy, sensitivity, and specificity of 0.65, 0.42, and 0.89, respectively [21]. Pattarabanjird et al. tested multiple ML models to predict anatomical CAD severity (extent of diameter stenosis) in a binary fashion using clinical variables along with SNP loci. Quantitative coronary angiography and the Gensini score, which is a summation score that quantifies the severity of CAD by accounting for the segment-based most severe stenosis and the location of the stenosis within the coronary arteries, were used to assess model performance. The best-performing model (Sequential Neural Network; training set *n* = 325 and internal validation set *n* = 82) accurately classified CAD severity with AUC of 0.84 in the validation set [22]. Similarly, Naushad et al. trained ML models to predict the presence of CAD and the percentage of coronary diameter stenosis using clinical and genetic variables. The best-performing model (an ensemble model; training set *n* = 648) accurately predicted CAD using 11 variables (clinical and genetic variants) with an AUC of 0.96 in the training set. The model also predicted the percentage of diameter stenosis with a correlation of 82.5% with the actual stenosis assessed using the gold-standard invasive angiography. However, these models were not internally nor externally validated [23]. 

Finally, the coronary artery calcium (CAC) score, calculated using the Agatston method on noncontrast ECG-gated cardiac computed tomography, is an established strong predictor of major adverse cardiovascular events in asymptomatic individuals. Genomic studies have previously focused on identifying genetic loci linked to CAC [24,25]. Oguz et al. suggested the use of ML algorithms to predict advanced CAC from SNP arrays and clinical variables. They identified a set of SNPs that ranked the highest in predictive importance and correlated with advanced CAC scores, defined as the 89th–99th percentile CAC scores in the derivation and replication cohorts, and trained different RF models to predict advanced CAC scores using clinical and genetic variables. Adding SNPs to clinical variables significantly improved AUC from 0.61 ± 0.02 to 0.83 ± 0.01; (*p* < 0.001) [26] for prediction of advanced CAC scores.

## 3. Risk Prediction Models and Imaging Modalities for Estimating Pretest Probability of CAD

Traditionally, stratifying patients presenting with stable chest pain using pretest probability (PTP) estimates of CAD has been commonly used to help with decision-making regarding downstream testing and the choice of an appropriate diagnostic modality. Historically, the Diamond–Forrester model—developed using age, sex, and chest pain characteristics—was used as a clinician’s risk stratification tool in predicting the PTP of CAD [27]. However, numerous studies showed its limitation in overestimating PTP by approximately threefold, especially in women [28]. This led to the development of the updated Diamond–Forrester model (UDF) and the CAD consortium score [29,30,31]. These scores, incorporating demographic and clinical risk factors, have been proven to be better at predicting the risk of CAD. Therefore, improving the ability to predict CAD using more accurate risk-assessment modeling is imperative, given the potential to reduce downstream testing and associated costs. Using clinical and demographic features, ML models have been employed to estimate the PTP of CAD [32,33,34]. In a recent multicenter cross-sectional study, a deep neural network algorithm based on the facial profile of individuals was able to achieve a higher performance than traditional risk scores in predicting PTP of CAD (AUC for the ML model 0.730 vs. 0.623 for Diamond –Forrester and 0.652 for the CAD consortium, *p* < 0.001) [35]. Though the study is limited by the lack of external validity and low specificity (54%), such approaches can potentially lead to a paradigm change in CAD management by facilitating earlier detection and initiation of primary prevention using readily available parameters, such as an individual’s facial profile.

When available, a CAC score has been shown to add to the PTP of CAD, with a CAC score of zero identifying low-risk patients who might not need additional testing [7,36]. ML models, combining clinical and imaging parameters, have been shown to have higher predictive power than traditional risk scores when predicting the PTP of obstructive CAD [37,38]. Al’Aref et al. included 25 clinical and demographic features to devise a ML model which, when combined with the Agatston CAC score, fared better than the ML model or CAD consortium score alone or in combination with the CAC score (AUC 0.881 for ML + CAC as compared to 0.866, 0.773, and 0.734 for the CAD consortium + CAC, ML model, and CAD consortium respectively, *p* < 0.05) [38]. As expected, CAC, age, and gender were the highest-ranked features in the model (Figure 1).

Various ML algorithms based on stress imaging, particularly single-photon emission computed tomography (SPECT), have been devised to facilitate the prediction of CAD. These models combined the clinical and demographic characteristics with the quantitative variables, as evaluated via SPECT to better predict CAD compared with the visual interpretation or quantitative variables alone [39,40,41,42,43,44]. More details about the parameters used to develop these models have been provided in Section 4, and a summary of the study results is included in Table 1.

Cardiac phase-space analysis is a novel noninvasive diagnostic platform that combines advanced disciplines of mathematics and physics with ML [45]. Thoracic orthogonal voltage gradient (OVG) signals from a patient are evaluated by cardiac phase-space analysis to quantify physiological and mathematical features associated with CAD. The analysis is performed at the point of care without the need for a change in physiologic status or radiation. Initial multicenter results suggest that resting cardiac phase-space analysis may have comparable diagnostic utility to functional tests currently used to assess CAD [46].

Finally, the assessment of regional wall motion abnormalities (RWMAs) on echocardiography has been associated with the presence of obstructive CAD, and as such can be useful in helping clinicians with downstream decision-making [47]. Recently, a deep-learning model developed by Kusunose et al. achieved performance similar to that of experienced cardiologists in the assessment of RWMAs on echocardiography (AUC of 0.99 vs. 0.98, *p* = 0.15) [48]. Other than the assessment of obstructive CAD, machine learning has found its wide applicability in echocardiography to predict ventricular capacities, abnormal valvular function, as well as cardiac hemodynamics, the discussion of which is outside the scope of this review paper [49,50,51].

## 4. Artificial Intelligence in Management of CAD in the Emergency Department

Chest pain is a common emergency department presentation, and distinguishing cardiac from noncardiac pain causes is crucial for optimal management. Modalities such as electrocardiography (ECG) serve as a quick way to recognize patterns associated with unstable CAD, and in particular acute coronary syndromes (ACSs). Deep neural networks have shown a consistent performance in image recognition, and models have hence been devised to identify patterns related to CAD and myocardial infarction (MI) [52,53,54]. By reducing interobserver variability and providing accurate results efficiently, this approach holds the promise of improving workflow across healthcare systems, while helping patients in areas of limited medical infrastructure and specialized care.

Cardiac biomarkers, such as high-sensitivity troponin, have been well-validated as markers of myocardial ischemia and damage [55]. High-sensitivity troponin I (hs-cTnI) assay forms the core of the ‘rule in and rule out’ clinical decision pathway as per ESC 2020 chest pain guidelines and 2021 ACC/AHA chest pain guidelines [36,56]. For instance, a very low hs-cTnI at hospital admission or a negative one-hour delta troponin (in the background of a low hs-cTnI value at admission) has a high negative predictive value (>99%) for ACS [57,58,59,60,61]. On the other hand, a high admission hs-cTnI value or a significant increase in values in an hour portends a high positive-predictive value (70–75%), warranting additional downstream testing [56,62,63].

Using the strategy mentioned above, approximately one-third of the patients fall in the ‘indeterminate’ zone. Diagnosis and management of this group is challenging, necessitating an approach based on clinical history, pre-existing risk factors, serial hs-cTnI trends, and further imaging. A recent ML model based on three clinical (age, sex, and prior percutaneous coronary intervention) as well as levels of three biomarkers (hs-cTnI, KIM-1, and adiponectin) demonstrated excellence in predicting obstructive CAD in the validation cohort (AUC 0.86 for prediction of >50% diameter stenosis) [64]. Notably, the model performed remarkably well in patients in the ‘indeterminate’ zone, with AUC of 0.88 and a positive predictive value of 93%, hence identifying patients who will benefit from further testing.

The 2021 American College of Cardiology/American Heart Association (ACC/AHA) chest pain guidelines advocate for the use of coronary CT angiography (CCTA) in intermediate-risk patients presenting with acute chest pain who either have no known history or a history of nonobstructive CAD (defined as coronary artery disease with less than 50% diameter stenosis) [36]. Given the ability of CCTA to accurately define coronary anatomy and extent/distribution of atherosclerotic plaque, it has been consistently shown to be a useful noninvasive imaging modality for patient selection, particularly for those who might require further invasive evaluation. However, interpretation of CCTA scans requires expertise and is time-intensive. Therefore, automatic interpretation of CCTA, which can lead to a significant reduction in the processing times, is highly desirable. ML algorithms have recently been developed, achieving a 70–75% reduction in reading time compared to that required for human interpretation (2.3 min for AI vs. 7.6–9.6 min for human readers). Though the model described performed slightly lower than highly experienced readers in interpreting CCTA (AUC 0.93 vs. 0.90 for human vs. AI, *p* < 0.05), when combined with low-experience human readers, it augmented the reader’s ability to correctly reclassify obstructive CAD (per-vessel net reclassification index (NRI) 0.07, *p* < 0.001) [65]. In addition, ML has been applied for various segmentation and classification tasks on cardiac CT imaging, from automatic segmentation of calcified and noncalcified plaque to automated calculation of the Agatston CAC score, and finally quantification of cardiac structures on CT imaging (Figure 2) [66,67,68,69,70,71,72,73]. Therefore, the application of ML could provide reliable results in real time, while bridging the dearth of experts in low-resource settings.

Stress testing, which provides an estimate of myocardial perfusion and viability, has been recommended as an alternative to CCTA in intermediate-risk chest pain patients [36]. Myocardial perfusion imaging, particularly SPECT, has been employed to recognize patients who might need an invasive evaluation, with a diagnostic sensitivity of 75–88% and specificity of 60–79% [74,75,76,77,78,79]. SPECT can be evaluated qualitatively in terms of size, severity, location, and reversibility of perfusion defect, and quantitatively, in terms of total perfusion deficit (TPD), summed stress score (SSS), summed rest score (SRS), as well as stress and rest volumes [80]. Automatically generated polar maps (representing radiotracer distribution in a two-dimensional plane) after three-dimensional segmentation of the left ventricle (LV) have been used as raw data for quantitative analysis. After the LV polar map is divided into 17 segments, each of the segments is graded on a scale of 0–4 based on the severity of ischemia. The scores are then summated to generate SSS and SRS [81]. Polar maps also provide information about the overall extent and magnitude of ischemia, in terms of TPD [81,82]. These objective variables extracted from the quantitative analysis offer an increased degree of reproducibility and can be incorporated into risk scores to predict mortality [82,83]. The diagnostic accuracy of qualitative and quantitative approaches is comparable, as has been shown in numerous studies [84]. A deep convolutional neural network-based model derived from polar maps (Figure 3) had a superior performance compared to TPD in predicting obstructive coronary artery disease (the AUC for ML were 0.80 and 0.76 vs. 0.78 and 0.73 for TPD on a per-patient and per-vessel basis respectively, *p* < 0.01). In addition to diagnosis, models to predict early revascularization (<90 days from SPECT) have been developed and have demonstrated better performance than individual SPECT variables on a per-patient and a per-vessel level [85,86].

## 5. Artificial Intelligence to Predict Functionally Obstructive CAD and Lesion-Specific Ischemia—As a Gatekeeper to the Catheterization Laboratory

One of the inherent limitations of CCTA is its limited ability to predict the functional significance of coronary stenosis. To overcome this shortcoming, CT-derived fractional flow reserve (FFR_CT_) was developed based on the critical concept of computational fluid dynamics (CFD), with numerous trials demonstrating its strong correlation with invasive fractional flow reserve (FFR) as determined by invasive coronary angiography (ICA) [87,88,89,90]. Rabbat et al. demonstrated that FFR_CT_ added to CCTA safely deferred ICA in patients with CAD of indeterminate hemodynamic significance. In addition, a high proportion of those who underwent ICA were revascularized [91]. These studies and others led to FFR_CT_ being incorporated in the 2021 ACC/AHA chest pain guidelines in intermediate-risk patients to detect lesion-specific ischemia in proximal or middle segments of the coronary arteries and determined to have atherosclerotic plaque with 40% to 90% diameter stenosis [36]. Despite its excellent correlation, the off-site computation of FFR_CT_ hampers its use in real time, owing to the need for longer processing times [92]. To overcome this limitation and to allow for quick computation of a value for the functional significance of a particular lesion, novel ML approaches based on artery lumen segmentation [93], left ventricular myocardial segmentation [94,95], and artery centerline tracking [96], have been proposed.

### 5.1. ML-Based CT-FFR Estimation and Diagnostic Accuracy

Based on the concept of artery lumen segmentation, the ML-based FFR estimation (CT-FFR_ML_) has generated significant interest in the past few years. The CT-FFR_ML_ model was trained on 12,000 synthetically generated coronary geometric datasets and used deep neural networks, allowing for automatic computation of FFR in real-time [93]. Coenen et al. performed a multicenter, prospective study to evaluate the diagnostic performance of CT-FFR_ML_ to predict lesion-specific ischemia, comparing it with traditional CCTA parameters, with invasive FFR being the gold standard [97]. They demonstrated an excellent correlation between CT-FFR_ML_ and FFR_CT_ (r = 0.997) and a superior performance of CT-FFR_ML_ over traditional CCTA in predicting lesion-specific ischemia (AUC: 0.84 vs. 0.69, *p* < 0.001 on a per-vessel level). Since then, multiple retrospective studies have been performed to evaluate the diagnostic accuracy of CT-FFR_ML_, validated against the gold-standard invasive FFR. They have further demonstrated superior diagnostic performance of CT-FFR_ML_ over CTA stenosis severity and quantitative atherosclerotic plaque features derived from CCTA [70,93,97,98,99,100,101,102,103,104,105,106,107].

To further highlight the incremental diagnostic value of CT-FFR_ML_ over anatomic plaque features derived from CCTA in vessels with intermediate stenosis, several other studies have been performed [99,102,103,106]. Tang et al. evaluated the diagnostic value of CT-FFR_ML_ in predicting lesion-specific ischemia [103]. Based on a study sample of 122 vessels in 101 patients, CT-FFR_ML_ performed better than anatomic CCTA parameters (AUC 0.96 for CT-FFR_ML_ vs. 0.63 for CCTA on a per-vessel basis *p* < 0.05).

### 5.2. Impact of Calcification Burden on the Performance of CT-FFR_ML_

The impact of coronary calcification on the diagnostic performance of CCTA has been well-established, with more extensive calcification limiting the ability of CCTA to evaluate for the presence of obstructive CAD [108,109,110]. Multiple indices have been devised to compute a CAC score, with the Agatston score, calcium volume, calcification remodeling index (CRI), and segmental arc calcification method being common examples [111]. The Agatston Score (AS) is the most widely validated approach, which summates the calcium score (function of peak density and area of the lesion) of the individual lesions across all coronary artery segments [112]. CRI provides a lesion-specific calcium estimate and is calculated as a ratio of the cross-sectional luminal area of the most severely calcified site to the proximal luminal area [113]. The segmental arc calcification method estimates lesion-specific calcium burden by measuring the greatest circumferential extent of coronary calcium, grading as nil (noncalcified), mild (0–90°), moderate (90–180°), and severe (>180°) calcification [110,114]. Recent studies have evaluated the performance of CT-FFR_ML_ with varying calcification burden as assessed by the parameters mentioned above [97,98,99,104]. Tesche et al. did a retrospective analysis using 482 vessels in 314 patients to evaluate the impact of calcifications on the performance of CT-FFR_ML_ [104]. They showed a statistically significant decrease in discriminatory power of CT-FFR_ML,_ measured in terms of AUC with increasing Agatston scores (AUC for CT-FFR_ML_ 0.85 and 0.81 in low–intermediate Agatston score (1–400) and high Agatston score (>400) ranges respectively, *p* = 0.04).

Di Jiang et al. [98] evaluated the impact of calcification arc and CRI on the performance of CT-FFR_ML_. No statistically significant difference was found in the discriminatory power of CT-FFR_ML_ with increasing calcification burden. In the proportion of patients where the Agatston score was available, there was no difference in the diagnostic performance of CT-FFR_ML_ across severity of calcification. The difference from Tesche et al. can be explained by a lower mean Agatston score (288 vs. 492 and 138 vs. 187 at a per-patient and per-vessel level, respectively) and smaller sample size (*n* = 150) for whom the Agatston score was available, resulting in low power to detect a difference.

Furthermore, Koo et al. [99] carried out a similar study and found no impact of increasing Agatston score on the performance of CT-FFR_ML_. Interestingly, a sizeable proportion of the sample had higher coronary calcification (mean Agatston score of 311 on a per-vessel basis). More research in this area is needed in order to further validate the diagnostic performance of CT-FFR_ML_ across varying degrees of coronary calcification.

### 5.3. CT-FFR_ML_ in Predicting Revascularization Events

CT-FFR_ML_ has been shown to be a better predictor than plaque features derived from CCTA for the determination of the presence of lesion-specific ischemia, but whether CT-FFR_ML_ influences the eventual treatment plan and outcomes (as guided by ICA-FFR) remains an active area of investigation [115,116,117,118]. Qiao et al. demonstrated the added benefit of CT-FFR_ML_ compared to relying on an anatomy-based strategy in patients with stable chest pain (reduction rate of ICA by 54.5% and 4.4% fewer revascularizations) [115]. Additionally, this study demonstrated that adding CT-FFR_ML_ to CCTA can decrease the rate of unnecessary ICA by 35.2% (thereby increasing the proportion of revascularizations when ICA is undertaken), truly acting as a gatekeeper to ICA. Furthermore, lower CT-FFR_ML_ was associated with higher major adverse cardiovascular event (MACE) risk when compared to diameter stenosis on CCTA (HR, 6.84 vs. 1.47) or ICA (HR, 6.84 vs. 1.84). Liu et al. found a similar rate of MACE (2.9%) after revascularization based on either combining CCTA stenosis ≥ 50% and CT-FFR_ML_ ≤ 0.8 or ICA stenosis ≥ 75% in a 2-year follow-up [116]. This study further highlighted the use of CT-FFR_ML_ as a gatekeeper to ICA with a positive impact on lower healthcare costs.

CT-FFR_ML_ comes with its own set of shortcomings. The diagnostic performance of the CT-FFR_ML_ model is lower, with the invasive FFR closely approaching the diagnostic threshold of 0.8 [97,99,119]. Traditional statistical and DL approaches have shown that stenosis severity; plaque characteristics, such as low-density, noncalcified plaque; and remodeling index are independent predictors of lesion-specific ischemia that are not related to CT-FFR_ML_ [120,121]. An integrated DL approach in the future that combines clinical features, anatomical plaque characteristics, vessel features, and functional assessment could potentially overcome this limitation.

## 6. Artificial Intelligence in the Field of Intracoronary Imaging

During ICA, intravascular ultrasound (IVUS) and optical coherence tomography (OCT) have been widely adopted for coronary luminal imaging, and some of the main applications involve assessment of plaque burden and optimization of stent placement [122]. IVUS uses ultrasound waves to generate cross-sectional images of coronary vessels with axial and lateral resolution ranging from 70–200 microns and 200–400 microns, respectively [122,123,124]. The penetration depth of IVUS is 10 mm, which allows for a complete cross-sectional analysis of the coronary vessel walls [124]. IVUS can help describe plaque characteristics, with high-risk plaques (plaques with large necrotic cores) appearing as areas of echo-attenuation [125]. On the other hand, calcifications in the IVUS frame indicate a calcified plaque, with heavily calcified plaque increasing the risk of stent underexpansion during percutaneous coronary intervention (PCI) [126,127]. Virtual histology IVUS (VH-IVUS) is another technique derived from radiofrequency data from IVUS, allowing for in vivo assessment of plaque composition [128]. By characterizing plaque features and vessel dimensions, IVUS has found its pre-procedural role in the quantitative and qualitative assessment of atherosclerotic plaque as well as interventional planning, ranging from vessel dimension assessment and evaluation of stent placement. Post-procedurally, IVUS can be employed to visualize stent expansion, identify stent edge dissection, stent mal-apposition, and confirm the presence of in-stent thrombosis in the right clinical context [129,130]. Given the benefits, IVUS has been shown to optimize stent implantation and improve outcomes, including revascularization, MACE, and mortality when used routinely in the cardiac catheterization laboratory [130,131,132].

On the other hand, OCT works on the principle of near-infrared light waves, generating cross-sectional images with a much higher axial and lateral resolution of 10 microns and 20–40 microns, respectively [133]. This allows for a detailed view of the lumen–plaque interface, providing accurate dimensions of the luminal area and better plaque characterization. The vulnerability of a plaque is a function of the thickness of its fibrous cap, the size of the necrotic core, and the presence of macrophages. A thin, fibrous cap; sizeable necrotic core; and increased macrophages increase the risk of plaque rupture and subsequent ACS [113]. Given the high resolution provided by OCT, it is considered a gold-standard invasive imaging modality for detecting thin-cap fibroatheroma (TCFA), which, pathologically, is a precursor of vulnerable plaque and clinically proven to be an independent predictor of MACE [134]. A significant drawback of OCT is its inherent low penetration depth (1–2 mm), which makes IVUS a better modality for a full-thickness analysis of vessel wall [130].

Though fascinating, IVUS and OCT have a low adoption rate in the US, being employed only at tertiary-care centers owing to cost, need for additional procedural time, and the associated technical complexities [135,136]. By using deep-learning algorithms to optimize the workflow associated with image acquisition and interpretation, ML has the potential to reduce procedural costs and time required, which are the two major hindrances to the widespread use of IVUS and OCT.

### 6.1. Artificial Intelligence to Optimize Peri-Intervention Workflow

To predict OCT-derived TCFA on IVUS images, Bae et al. created a ML model, enrolling 517 patients who underwent ICA [137]. A total of 40,908 IVUS-OCT co-registered sections in 517 coronary arteries were divided into training and testing sets in a ratio of 4:1. An artificial-neural-network-based model using 17 features achieved the highest performance with a sensitivity and specificity of 85 ± 4% and 79 ± 6%, respectively, and good discriminatory power (AUC of 0.80 ± 0.08). Larger plaque burden, minimal diameter, decreased lumen area, and increased lumen eccentricity were seen to be strongly associated with OCT-derived TCFAs. Min et al. utilized a deep learning algorithm (densely connected convolutional neural network) on 35,678 OCT frames to automatically detect TCFAs from OCT images [138]. After the frames were interpreted for the presence/absence of TCFA, data was fed into the algorithm to devise a deep-learning model. By achieving high sensitivity and specificity of 88.7 ± 3.4% and 91.8 ± 2.0% on the test data, such deep-learning models can significantly reduce processing times and allow for easy interpretation when it comes to identifying a vulnerable high-risk plaque.

As mentioned earlier, IVUS can help characterize high-risk plaques, which appear as areas of attenuation on IVUS frames due to the presence of a large necrotic core. Identifying such lesions becomes imperative to reduce the incidence of complications such as periprocedural MI. To accurately classify plaque characteristics and to facilitate detection of high-risk lesions, Cho et al. described a deep-learning algorithm to accurately differentiate IVUS segments as attenuated or calcified, or plaque without attenuation or calcification [139]. A total 598 vessels in 598 patients were evaluated, and a DL model with five-fold cross-validation was developed. The deep-learning model closely correlated with the expert read, and correlation coefficients for calcification, attenuation, and no attenuation or calcification were 0.79, 0.74, and 0.99, respectively (Figure 4).

Stent underexpansion is a frequently encountered entity that has been associated with an increased risk of in-stent restenosis. Studies have demonstrated the postprocedural minimum stent area (MSA) and IVUS-measured stent length to be independent predictors for in-stent restenosis [130,140,141,142,143]. Min et al. devised a deep-learning model to predict stent underexpansion based on pre-PCI IVUS frames [144]. They evaluated 618 coronary lesions from 618 patients undergoing pre- and postprocedural IVUS and divided them into training and testing sets in a 5:1 ratio. A convolutional neural network (CNN) model was used to predict the poststenting stent area. Features extracted from the CNN were combined with additional image-derived features via a boosted ensemble algorithm, which yielded sensitivity and specificity of 68% and 98%, respectively, and an AUC of 0.95 to predict stent underexpansion. The stent areas and volumes predicted via the CNN correlated well with poststenting IVUS (r for stent area and volume 0.832 and 0.958, respectively). The most important features predicting stent underexpansion were luminal area, external elastic membrane (EEM) area (both at the reference and the target), and plaque area of the region of interest.

### 6.2. Applications of Artificial Intelligence in Intra and Post-Intervention Workflow

Optimal stent expansion is vital to successful outcomes, with stent underexpansion predisposing to stent restenosis and a greater stent expansion exposing the procedure to a risk of stent edge dissection [130]. IVUS, by allowing direct visualization of vessel architecture, can help in the earlier identification and management of these complications. Nishi et al. developed a ML model to compute the luminal area and the vessel area accurately, as well as the stent area, which exhibited an excellent correlation between ML-derived and expert-derived dimensions while dramatically reducing the time required for segmentation of IVUS images (37 s) compared with expert analysis (30 h) [145].

Virtual histology IVUS (VH-IVUS) is a well-studied intracoronary imaging modality used for in vivo visualization of high-risk plaques [146,147,148,149]. Zhang et al. devised a deep-learning model to predict the location of high-risk plaques in nonculprit vessels in patients who underwent IVUS at baseline and after one year [150]. Though large-scale validation is required, the model predicted the occurrence of TCFAs, plaque burden >70%, and minimal luminal area ≤4 mm^2^ reasonably well at a one-year follow-up on a per-lesion level.

## 7. Artificial Intelligence-Based Post-Procedure Risk Prediction Models

In addition to early detection and the institution of guideline-directed therapy in the appropriate risk strata, accurate prediction of unheralded adverse events forms the cornerstone for managing CAD. Identifying the high-risk target population can potentially provide a window for aggressive risk factor modulation, thereby reducing mortality and contributing towards better health at a population level. Multiple risk-prediction models have been developed to predict in-hospital mortality and the long-term risk of MACE in high-risk cohorts [151,152,153,154,155,156,157,158].

PCI is a relatively safe procedure, with a reported overall in-hospital mortality rate of 1–2% [159]. The risk of complications increases with increasing patient morbidity, with an incidence of technical difficulties and periprocedural complications 2.2 times higher than in the average population [160]. The Mayo clinic risk score (MCRS) and New York State risk score (NYSRS) were developed to predict in-hospital and 30-day mortality in patients undergoing PCI. Both scores performed equivalently well, showing an excellent discriminative ability to identify patients at a higher risk for in-hospital and 30-day mortality [161]. They employed regression-based models, assuming a linear interplay between patient variables and mortality outcomes. ML models have been recently developed to potentially uncover complex and nonlinear relationships between multiple factors, hence improving diagnostic accuracy over current models.

Zack et al. evaluated 11,709 patients to train two RF regression models—one using 52 demographic and clinical parameters to predict in-hospital mortality and the second model also incorporating 358 discharge variables in addition to the 52 admission parameters to predict 180-day cardiovascular mortality and 30-day heart failure rehospitalization [162]. They compared the model performances against logistic regression models trained using the same variables. No significant difference was found between the RF model and logistic regression in predicting in-hospital mortality (AUC 0.923 vs. 0.925, *p* = 0.84). The ML model performed significantly better than the logistic regression model for prediction of 30-day heart failure hospitalizations (AUC 0.899 vs. 0.846, *p* = 0.003) and 180-day cardiovascular death (AUC 0.881 vs. 0.812, *p* = 0.02).

Al’Aref et al. [163] developed a supervised machine learning approach to predict in-hospital mortality among patients undergoing PCI. Utilizing 479,804 patients from the New York state registry, they utilized 49 clinical, angiographic, and periprocedural event characteristics to create a ML model via adaptive boosting. It performed better than the logistic regression model (AUC 0.927 for ML vs. 0.908 for logistic regression, *p* < 0.01). Age and ejection fraction emerged as the most important variables predicting mortality.

Periprocedural bleeding is one of the most common complications of PCI and has been linked to adverse in-hospital outcomes [164,165]. Current risk scores such as the NCDR bleeding risk-prediction model and the simplified NCDR bleeding-risk score have performed modestly well in identifying patients at a high risk of periprocedural bleeding [166]. To improve the performance of the existing risk model, an ML-based model was developed on 3,316,465 patients enrolled in the CathPCI registry [167]. In addition to the 31 variables used in the existing model, 28 new variables were incorporated to devise an integrated model via the gradient-boosting approach. The blended model using ML had a higher discriminatory power than the existing model (*C* statistic 0.82 vs. 0.78, *p* < 0.05) and improved the positive predictive value to 26.6%, compared with 21.5% for the existent model.

One of the primary challenges faced in the PCI era is in-stent restenosis, which is linked to neointimal proliferation due to vascular wall damage [168]. The incidence of ISR has been estimated to be 20–40% for bare metallic stents and 10–15% for drug-eluting stents [168,169]. Smaller vessel size, increasing stent length, complex lesion morphology, diabetes mellitus, and prior bypass surgery are risk factors for stent restenosis [169]. These factors have been incorporated with other variables to devise risk models such as PRESTO 1, PRESTO 2, and EVENT scores to provide an estimated risk of ISR [170,171]. These models have a modest discriminatory power in predicting ISR, leaving room for improvement. A big-data approach incorporated 68 variables relating to clinical, demographic, and angiographic characteristics to devise a risk prediction model for ISR [172]. The ML model, when applied post-PCI, achieved a higher discriminatory power (AUC for the precision recall curve was 0.45 vs. 0.31, 0.27, and 0.18 for PRESTO-1, PRESTO-2, and EVENT, respectively, *p* < 0.05) to predict ISR at 12 months. Interestingly, post-PCI TIMI flow was one of the prominent predictors of ISR, alongside diabetes mellitus and the presence of ≥2 vessel CAD. Though the model requires external validation, given the small sample size of the population (*n* = 263), the study yet again underscores the merit of ML in identifying crucial parameters from a vast dataset to predict outcomes.

## 8. Artificial Intelligence-Based Long-Term Mortality and MACE Prediction Models

Prognostic modeling via ML has been validated with the use of electronic health records (EHRs) integrated with clinical scores and imaging modalities to predict MACE [173,174,175]. Utilizing the array of data available in EMR and identifying patterns based on clinical course, ML models have been used to create a personalized treatment algorithm (ML4CAD) for every patient, based on risk factors, past medical history, time present in the EMR system, and medications. The illustrated model makes clinical decisions for patients based on these factors and suggests a decision with an aim to increase prescription effectiveness, evaluated in the terms of time from initial diagnosis to the first potential adverse event (time to adverse event, TAE). The model had superior performance when compared to standard of care, increasing the time to adverse event (TAE) from 4.56 to 5.66 years (24.3% increase), hence furthering the idea of precision medicine [174,176].

Imaging findings, such as CAC score quantified from cardiac computed tomography, are an independent risk factor adding to the traditional clinical risk factors in predicting long-term risk of cardiovascular events [177,178,179]. Noncontrast CT imaging, other than providing information on the CAC score, provides valuable measures such as epicardial adipose tissue (EAT) volume, and EAT attenuation, all of which have been shown to provide additional information regarding the long-term risk of cardiovascular disease [180,181,182]. Extracting these pieces of data can be tedious and labor-intensive, and automated techniques can result in more standardized evaluations in a more time-efficient manner.

Multiple ML techniques have been proposed to automatically evaluate CAC score from dedicated cardiac and non-EKG gated chest CT scans [66,67,183,184,185]. ML techniques incorporating CAC score and other imaging parameters have been shown to be a better predictor than the traditional risk scores employed for cardiovascular disease risk stratification [181,186,187,188,189]. An ensemble-boosting model developed by Nakanishi et al. incorporating a total of 77 clinical and imaging variables had a superior discriminatory power for predicting coronary heart disease deaths than imaging and clinical data alone (AUC for ML model: 0.845 compared to 0.821 and 0.781 for clinical data and CAC respectively, *p* < 0.001) (Figure 5) [190].

Apart from CAC scoring and traditional CT metrics, the role of EAT volume and attenuation in the prediction of future cardiovascular risk has been an active area of research. Deep-learning approaches to automatically compute EAT volume and EAT attenuation from CT have been developed, significantly reducing generation time from 15 min to 2 s [186]. Eisenberg et al. demonstrated an independent association between deep-learning-derived EAT volume and attenuation with the risk of future MACE, defined as myocardial infarction, late (>180 days) revascularization, and cardiac death (HR:1.35, *p* < 0.01 and 0.83, *p* = 0.01, demonstrating a direct correlation with EAT volume and an inverse correlation with EAT attenuation respectively) [187]. Subsequently, these parameters have been combined with other physiologic and radiology variables to develop new deep-learning approaches, which have further been shown to have a higher predictive value than the traditional risk scores [186,189]. These have been summarized in Table 2.

Apart from its role in CAD diagnosis, CCTA has been shown to have an incremental prognostic value in terms of short- and long-term risk prediction. Results from the CONFIRM registry validated two CCTA parameters, namely the number of proximal segments with stenosis > 50% and the number of proximal segments with mixed or calcified plaque as important prognostic markers above the predictive value of the Framingham risk score (FRS) [191,192,193].

A multitude of ML approaches have been described, combining imaging parameters with clinical and demographic parameters for better prognostication of cardiovascular outcomes [194,195,196,197,198,199] Including 10,030 patients with suspected CAD from the CONFIRM registry, Motwani et al. utilized a boosting ensemble algorithm using 25 clinical and 44 CCTA parameters [195]. The ML algorithm performed better in predicting 5-year all-cause mortality than CCTA segment stenosis score or FRS (AUC 0.79 for ML vs. 0.664 for segment stenosis score and 0.61 for FRS, respectively, *p* < 0.001). More recently, models incorporating high-risk plaque features with the traditional imaging and clinical parameters have performed better than either of the parameters in isolation [196,197]. A review of literature summarizing all the studies is presented in Table 3.

Although anatomical CT scores and plaque features provide useful diagnostic and prognostic data, the complex interplay of factors at the molecular level, in addition to patient-level characteristics leading to specific phenotypic manifestations in terms of plaque burden and features, is not well-elucidated and remains an area of active research. In particular, elucidating important factors that “drive” the process of atherosclerotic plaque formation and progression is not only vital from a therapeutic perspective, but it can also improve risk-assessment strategies. Recent studies have demonstrated that coronary artery inflammation inhibits lipid accumulation in the perivascular adipose tissue [200]. This results in a higher attenuation of the affected perivascular area, identified on CCTA as the fat attenuation index (FAI). FAI has been shown to be a sensitive marker of coronary inflammation, with higher FAI values (≥−70.1 HU) independently predicting cardiovascular mortality [200,201]. A posthoc analysis of the CRISP-CT study showed an incremental value of adding FAI to high-risk plaque characteristics, pointing towards a more significant role of these precursor lesions in predicting patient outcomes [202]. A more recent ML approach created a pericoronary fat ‘radiomic’ profile (FRP), identifying radiomic variables predicting tissue inflammation, fibrosis, and vascularity on CCTA [203]. The incorporation of FRP significantly improved the MACE predictive ability of the traditional model (AUC for traditional + FRP 0.88 vs. 0.754 for the traditional model, *p* < 0.001). Using a cut-off of 0.63, individuals in the high FRP group were at a higher risk of MACE (HR = 10.84, *p* < 0.001). Importantly, Kaplan–Meir analysis showed an additional value of FRP over high-risk plaque (HRP) characteristics in predicting long-term survival (HR for the FRP-/HRP+ subgroup 5.97, *p* = 0.03 compared to 43.33 for the FRP+/HRP+ subgroup). Such ‘radiotranscriptomic’ approaches incorporating molecular biology and radiology and evaluating their interaction via artificial intelligence can help uncover deeper relationships between metabolic pathways and clinical outcomes, helping to better understand the pathophysiology and elements involved in the clinical progression of cardiovascular disease.

## 9. Discussions

With significant developments occurring in the last decade in terms of data processing and analytics, AI can provide new and sophisticated tools that could help us to better understand disease processes, which ultimately should translate into better patient care and outcomes (Figure 6). Nevertheless, AI comes with its own set of limitations. ML models lack interpretability and suffer from the ‘black box’ problem [204]. ML models based on neural networks and ensemble methods are inherently complex and are derived from complicated mathematical algorithms. ‘Explainable (interpretable) machine learning’, whereby simple approximations of the model are devised to make it more understandable, is being developed to overcome the black box problem [205,206].

Another limitation of ML encountered at the model-development phase is sampling bias and lack of external validation [207,208]. ML learning models usually derive their weights from large datasets. Datasets, particularly those derived from EHRs, might be skewed and not representative of the entire population, leading to significant sampling bias and limited generalizability. A few models have tried to address this problem by stratifying the datasets at the model-development phase to ensure not to lose representation of any subgroup and preserve the model’s generalizability. Nevertheless, randomized controlled trials are needed to potentially overcome this bias and establish the model performance against the standard clinical parameters. In addition, imputation methods such as MICE have been used to address the missing data issue [209].

Furthermore, the creation of bigger datasets by pooling data from multiple hospital systems has led to a lack of standardization of datasets, potentially compromising the quality of analysis. Datasets might internally differ from each other because of the different mechanisms used to generate them. For instance, one dataset might define the presence of diabetes mellitus through ICD-10 codes, while another dataset might define it using glycemic indices, such as the hemoglobin A1c. On a similar theme, ML models developed by using imaging modalities deserve a special mention. For instance, differences can exist at the level of image scanning (different scanner characteristics and vendors), image quality (radiation dose, motion artifacts), and image processing (reconstruction filters, post-processing) which can potentially lead to significant variability and differences of the assimilated data. A prerequisite to the development of any ML model is the centralization of data, which is tedious given the different image processing algorithms employed at various institutions. This lack of standardization needs to be addressed before AI can be fully integrated into clinical practice.

Overfitting is another concern encountered during ML model development, which occurs when the algorithm learns the data ‘too well’ and interprets the signal noise as concepts [210]. This usually happens with smaller datasets and can lead to a lack of external validity, despite high performance in the training and internal validation datasets. A definite solution is k-fold cross-validation, whereby data is randomly divided into an arbitrary *k* number of partitions. The model is trained using *k* − 1 number of data subsets and tested on the remaining subset. This process is repeated *k* total number of times, using different combinations of training and testing datasets to select the best model hyper-parameters to yield the final model. This can potentially reduce noise and lead to better generalizability of the model in the overall population.

Apart from the problems encountered at the model development and training phase, there are a few noteworthy practical limitations to the implementation of ML within healthcare workflows. Firstly, unauthorized data access is an issue, as handling such large amounts of data also poses a risk of leaking sensitive patient information, thereby violating patient confidentiality and privacy [211]. Furthermore, comparisons between various machine-learning methods are difficult, given the different combinations of model parameters and different population characteristics used for in model development. Hence, it becomes difficult for physicians to compare and choose one model over the other. Prospective future trials, comparing these models on the same dataset, are needed to select the best algorithm fit for integration into routine clinical decision-making. Proper integration of AI can only be achieved once these models are embedded within EHRs. However, the full implementation and assimilation of developed AI models into EHRs can be a complex issue, as it depends on organizational resources and patient-privacy policies. Furthermore, available algorithms may be limited to off-the-shelf ML models, rather than more intricate and complex neural networks, which is easier to implement in a real clinical setting. Yet, a data-driven approach utilizing advanced analytic techniques can help clinicians and patients to make informed decisions, improve care, and optimize workflow efficiency.

## 10. Conclusions

In conclusion, AI provides an unprecedented potential to transform healthcare and enhance the current system’s ability to serve populations at large, while providing tools to focus on individualized yet comprehensive and precise care.

## Figures and Tables

**Figure 1 healthcare-10-00232-f001:**
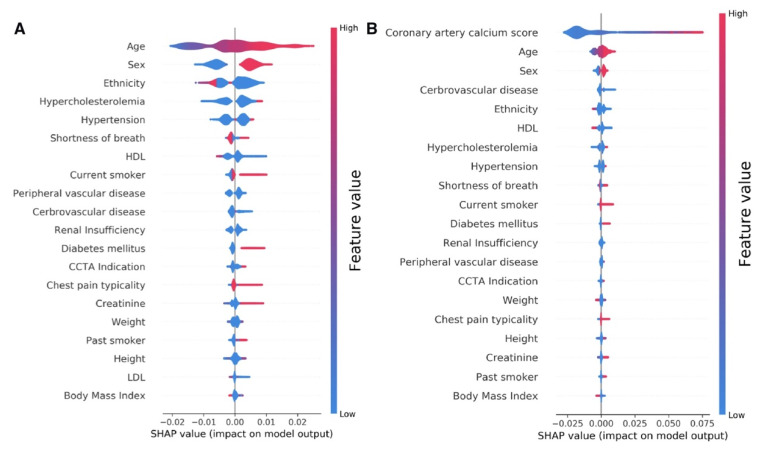
Feature ranking in the machine-learning model developed by Al’Aref et al. based on clinical and demographic factors (**A**) and when combined with the Agatston calcium score (**B**), for the prediction of the presence of obstructive CAD on coronary CT angiography. A more positive SHAP (Shapley additive explanation value) indicates higher importance of the variable in the machine-learning model. Adapted with permission from Al’Aref et al. [38], Oxford University Press.

**Figure 2 healthcare-10-00232-f002:**
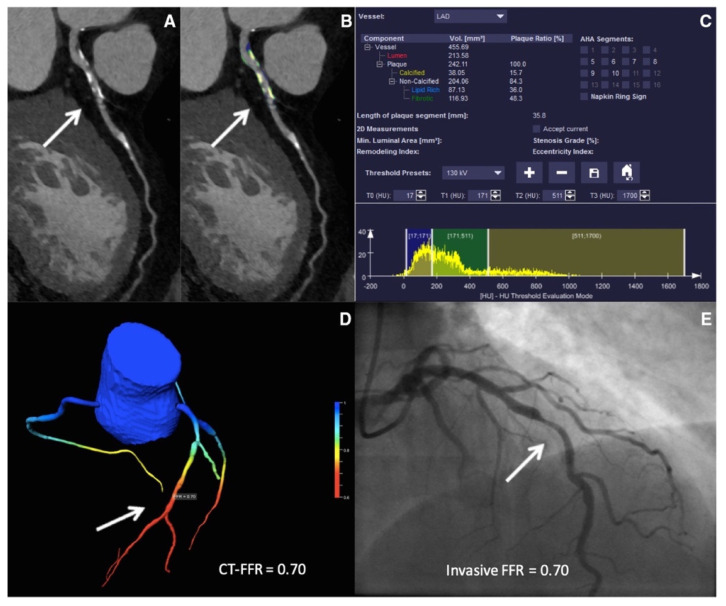
ML-based fractional flow reserve from cardiac CT (CT-FFR_ML_). Machine-learning-based coronary plaque analysis quantifies atherosclerotic plaque into calcified and noncalcified components (**A**,**B**). This is further integrated with other quantitative parameters (**C**) and transformed into 3-D images of the vessels to give CT-FFR_ML_ (**D**), which has been shown to have a good correlation with invasive fractional flow reserve (FFR—**E**). Adapted with permission from Von Knebel Doeberitz et al. [65], Elsevier.

**Figure 3 healthcare-10-00232-f003:**
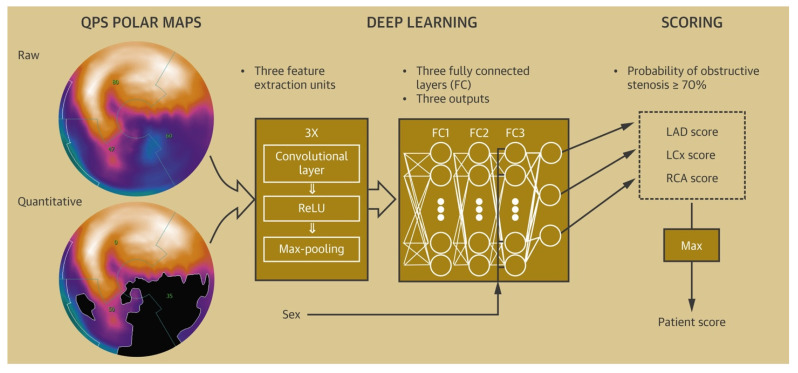
Deep-learning model to predict obstructive CAD from polar maps. Raw polar maps and extent polar maps (maps with abnormal pixels representing ischemia blackened out) are fed into deep neural networks, with the extracted data used to calculate scores for individual vessels to predict the probability of CAD. Adapted with permission from Betancur et al. [43], Elsevier.

**Figure 4 healthcare-10-00232-f004:**
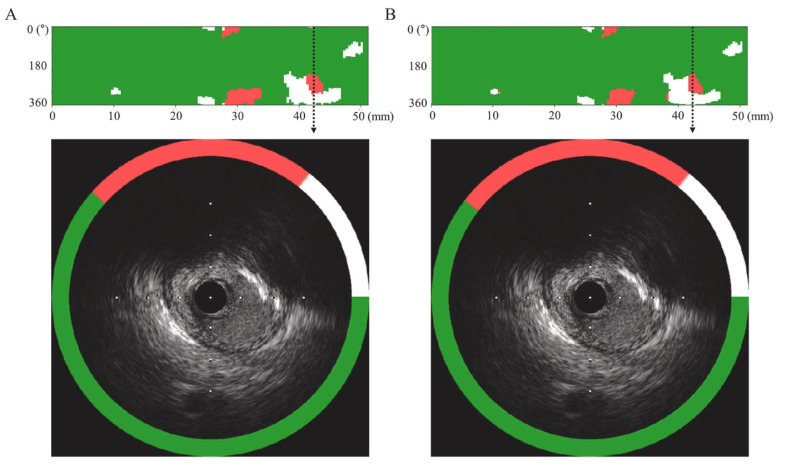
ML(**A**) vs. human (**B**) interpretations for plaque characterization for IVUS images. The upper panel shows representation of plaque features along the long axis of the vessel (*x*-axis represents the distance from ROI (region of interest) and *y*-axis represents the angular position (0–360°) of the plaque. The lower panel shows the plaque characterization on a cross-sectional view of the IVUS frame. Attenuation, calcification, and regions without attenuation or calcification are represented by red, white, and green respectively. Adapted with permission from Cho et al. [135], Elsevier.

**Figure 5 healthcare-10-00232-f005:**
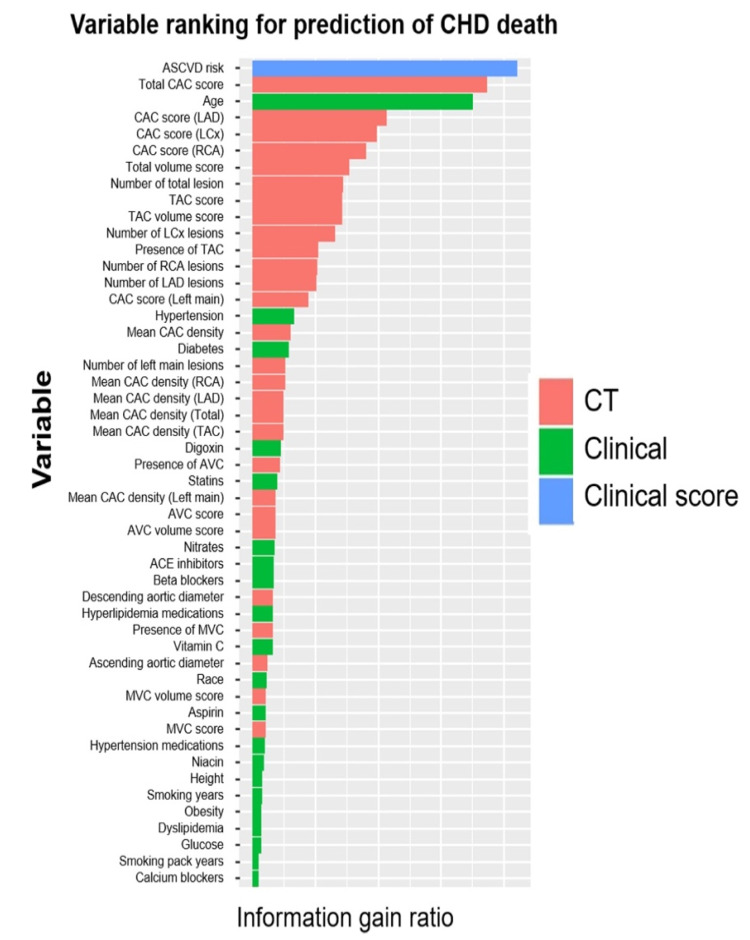
Variable importance as determined by the ML model for prediction of coronary heart disease deaths. Abbreviations: CAC: coronary artery calcium; TAC: thoracic aortic calcification; AVC: aortic valve calcification; MVC: mitral valve calcifications; LAD: left anterior descending; LCx: left circumflex RCA: right coronary artery. Adapted with permission from Nakanishi et al. [190], Elsevier.

**Figure 6 healthcare-10-00232-f006:**
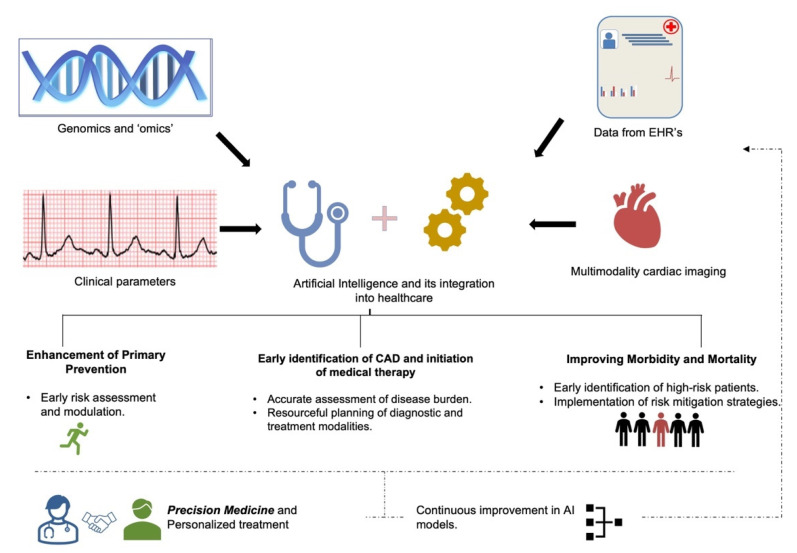
Current applicability and future directions for AI in coronary artery disease.

**Table 1 healthcare-10-00232-t001:** Studies comparing ML models developed using SPECT variables with those using the qualitative or quantitative variables for prediction of CAD.

Study	Center/Sample Size	ML Technology	Brief Description and Outcomes	Result	Limitations
Guner et al. [41] 2010	Retrospective Single-center study 243 patients	Artificial neural networks	ML model trained from image data from stress and difference (devised from rest and stress maps) polar maps. Outcome: ML model vs. expert interpretation in the prediction of obstructive (>70% stenosis) CAD	AUC 0.74 and 0.84 for ML and expert read, no statistical difference found between ML-trained model and expert read.	Small sample size Limited availability of software used.
Arsanjani et al. [44] 2013	Retrospective Single-center study 1181 patients	Boosted ensemble	ML model using quantitative variables (TPD, stress/rest perfusion change, TID) and clinical variables (age, sex, and post-ECG probability) created. Outcome: ML vs. visual analysis and TPD in prediction of obstructive CAD.	AUC: ML (quantitative + clinical − 0.94 ) > ML (quantitative, 0.90) > combined supine/prone TPD − 0.88. Also, better than experts (0.89 and 0.85 for two different experts).	Dual isotope imaging protocol used, leading to difficulty in comparing rest and stress images. No information was given on localization of ischemia (didn’t provide information about the culprit vessel).
Arsanjani et al. [39] 2013	Retrospective Single-center study 957 patients with no history of CAD.	Support vector machines	ML model using quantitative and functional variables derived from SPECT. Outcome: ML model vs. quantitative and visual analysis in prediction of obstructive CAD or LAD stenosis > 50%.	AUC: ML (0.92) > TPD (0.90) > Expert analysis (0.88 and 0.87 for two different experts)	Limited generalizability (patients with a history of CAD and valvular disease were excluded). Stenosis on CAG determined qualitatively rather than quantitatively.
Betancur et al. [43] 2018	Retrospective Multicenter study 1638 patients	Convolutional neural networks	DL model developed from single-view polar maps; trained and compared with TPD for prediction of CAD. Outcome: ML model vs. TPD for prediction of obstructive CAD.	DL > TPD on per patient (AUC 0.80 vs. 0.78) and per vessel level (AUC 0.76 vs. 0.73) for prediction of obstructive CAD, *p* < 0.01.	Stenosis on CAG determined qualitatively rather than quantitatively. Only stress static images used to train the algorithm.
Betancur et al. [40] 2018	Retrospective Multicenter study 1160 patients with no history of CAD	Convolutional neural networks	DL model developed to automatically combine upright and supine MPI polar maps. Outcome: ML model vs. TPD for prediction of obstructive CAD.	DL > TPD on per patient (AUC 0.81 vs. 0.78) and per vessel (AUC 0.77 vs. 0.73) for prediction of obstructive CAD, *p* < 0.001	Stenosis on CAG determined visually. Only stress MPI images were taken.
Rahmani et al. [42] 2019	Retrospective Single-center study 93 patients	Artificial neural networks	ML model created using clinical, demographic, and polar-map data. Outcome: ML model vs. expert interpretation in prediction of obstructive CAD and abnormal angiographic results.	Accuracy for ML vs. visual interpretation for prediction of: Obstructive CAD:85.7% vs. 65.0% Abnormal angiographic results: 92.9 % vs. 81.7%	Small sample size Patients with a high pretest probability included, hence possible over- and underestimation of sensitivity and specificity respectively.

CAG: coronary angiography; LAD: left anterior descending; MPI: myocardial perfusion imaging, TPD: total perfusion deficit, TID: transient ischemic dilation.

**Table 2 healthcare-10-00232-t002:** Studies evaluating the impact of coronary artery calcium score (CACS) among other variables in the prediction of mortality in patients with no history of coronary artery disease.

Study	Study Design/Sample Size	ML Model	Brief Description and Follow-Up	Results	Limitations
Eisenberg et al. [187] 2020	Prospective single-center study, 2068 asymptomatic patients	Convolutional neural network	To check for impact of EAT volume and EAT attenuation computed via deep learning in prediction of MACE, defined as defined as MI, late (>180 days) revascularization and cardiac death. Follow up: >14 years	Increased EAT volume (HR: 1.35) and decreased EAT attenuation (HR 0.83) independently associated with MACE in addition to CACS (HR 1.25) and ASCVD score (HR 1.03), *p* < 0.01 for all.	Study done on asymptomatic patients; external validation needed if applied on symptomatic patients. Previous-generation CT scanners used (data collected from 1998–2005).
Han et al. [188] 2020	Retrospective multicenter study, 86,155 asymptomatic patients	Boosted ensemble	ML model with 35 clinical, 32 lab, and 3 CACS parameters (CACS, calcium volume, and calcium mass) in prediction of all-cause mortality Median follow up: 4.6 years	ML (0.82) > ASCVD score + CACS (0.74) > Framingham risk score + CACS (0.70)—reported as AUC in the test set. No statistical difference in the performance in the validation set.	Retrospective All-cause mortality reported rather than specific cardiac endpoints.
Nakanishi et al. [190] 2021	Multicenter observational study, 66,636 asymptomatic patients	Boosted ensemble (Logitboost)	ML model incorporating 46 clinical and 31 CT variables—CAC score, extra coronary scores (not including EAT) in prediction of cardiovascular (CHD + stroke + CHF + other circulatory diseases), and coronary heart disease (CHD) deaths Follow up: 10 years	For cardiovascular deaths: AUC for ML (all) 0.845 > ASCVD (0.821) > CAC score (0.78). For coronary heart disease deaths: AUC for ML (all) 0.860 > ASCVD (0.835) > CAC score (0.816).	Multiple CT variables, including EAT, were not available for some patients.
Commandeur et al. [186] 2020	Prospective single-center study, 1912 asymptomatic patients	Boosted ensemble (XgBoost)	ML model using clinical variables, plasma lipid panel measurements, CAC, aortic calcium, and automated EAT measures in prediction of MI and cardiac deaths. Median follow up: 14.5 years	ML model 0.82 > ASCVD risk score 0.77 ~ CAC 0.77. Age, ASCVD risk score, and CACS were the three most important features seen in the model.	Overfitting; since small number of events (<4%). Study done on asymptomatic patients; external validation needed if applied on symptomatic patients.
Tamarappoo et al. [189] 2021	Prospective single-center study, 1069 asymptomatic patients	Boosted ensemble (XgBoost)	ML model using 12 variables from ASCVD score, 5 CT parameters (including EAT volume and attenuation) and top 15 serum biomarkers) to predict cardiac events Mean follow up: 14.5 years	ML (0.81) > CAC (0.75) > ASCVD (0.74).	Single-center study Overfitting; given the small number of cardiac events during follow up (~2%)

ASCVD: atherosclerotic cardiovascular disease; CHF: congestive heart failure; EAT: epicardial adipose tissue; HR: hazard ratio; MI: myocardial infarction.

**Table 3 healthcare-10-00232-t003:** Summary of literature regarding mortality outcomes using CCTA data.

Study	Study Design/Sample Size	ML	Brief Description and Outcomes	Results	Limitations
Motwani et al. [195] 2016	Multicenter prospective study, 10,030 patients with suspected CAD	Boosted ensemble (LogitBoost)	25 clinical and 44 CCTA parameter used to create ML model Outcome: Prediction of 5-year ACM; compared against clinical risk scores and CCTA parameters.	AUC: ML (0.79) > Segment stenosis score (SSS) (0.64) and FRS (0.61); *p* < 0.001.	Observational; concern for selection bias Cardiac-specific endpoints were not defined, given the data unavailability.
Hoshino et al. [198] 2016	Multicenter retrospective study, 220 patients with intermediate LAD stenosis	Unsupervised hierarchical clustering	Two clusters (CS1 and CS2) using 42 variables created via ML. Outcome: Relation between FAI and CCTA defined clusters, Prognostic value of ML-derived clusters in combination with FAI.	Age, CS1 features (higher plaque volume, remodeling index, higher FAI amongst others), and FAI were independent predictors of MACE. Improved NRI with (FRS + CS1 + FAI) as compared to FRS alone.	Retrospective, small size Majority of vessels were LAD; hence the study was restricted to a specific population. 40% cardiac events were non-LAD revascularization; hence the results were not generalizable.
Van Rosendael et al. [197] 2018	Multicenter prospective study, 8844 patients with suspected CAD	Boosted ensemble	35 variables (SS and plaque composition for 16 coronary segments and 3 additional variables) compared with traditional CT scores. Outcome: ML vs. traditional CT scores in predicting 5-year composite MI and death.	AUC for ML (0.77) > SSS (0.70)	No comparison with clinical risk scores Retrospective study with risk of selection bias
Johnson et al. [194] 2019	Single-center retrospective study, 6892 patients	K nearest neighbors	ML model (64 vessel-related features) vs. CAD-RADS. Outcome: Prediction of ACM, CAD-related deaths. Also, decision to start statin.	AUC for all-cause mortality (0.77) > CAD-RADS (0.72); AUC for CAD-related deaths—ML (0.85) > CAD-RADS (0.79). Significant increase in sensitivity with ML model.	Retrospective study with limited population diversity Unblinded CCTA results that might have affected event incidence
Johnson et al. [199] 2020	Single-center retrospective study, 6892 patients		ML model developed via radiologist report. Outcome: Prediction of ACM and CAD-related mortality; compared against FRS. Also, decision to start statin.	ACM: AUC for ML (0.85) > FRS (0.79) CAD related deaths: AUC for ML (0.87) > FRS (0.82) Using ML, equally high sensitivity but significant reduction in unnecessary statin prescription (AUC for ML 0.89 vs. FRS 0.75).	Retrospective study design Concern for misclassification bias due to incomplete follow-up
Tesche et al. [196] 2021	Single-center retrospective study, 361 patients with suspected and confirmed CAD	Boosted ensemble (RUSBoost)	28 clinical, CCTA scores and adverse plaque characteristics included. Outcome: 5-year MACE prediction; compared against FRS, CCTA scores and adverse plaque features.	AUC for ML (0.96) > AS (0.84) > FRS (0.76). Important imaging parameters: SSS, obstructive CAD of RCA. Important clinical factors: age, FRS	Small sample size, retrospective study design Follow-up using medical records No external validation to test prognostic accuracy

ACM: all-cause mortality; AS: Agatston score; CAD-RADS: coronary artery disease reporting and data system; CS: cluster sample; FAI: fat attenuation index; FRS: Framingham risk score; RCA: right coronary artery; SSS: segment stenosis score.

## Data Availability

Not applicable.

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
