# Peer review of "Current and Future Applications of Artificial Intelligence in Coronary Artery Disease"

_healthcare, 2022, doi:10.3390/healthcare10020232_

Round 1

Reviewer 1 Report

The paper of Gautam N et al on applications of Artificial intelligence in coronary artery disease is an excellent review paper , good writing and while complex issue , the authors have written clearly extesively all concepts on CAD and artificial intelligence.

Every paragraph was carefully documented with rich bibliography, about that I report ony one reference used, the n 13 , the authors are sure that is correct?

Ultimately , in my opinion this is an important review that I recommended to publish on Healthcare

Author Response

Dear Reviewer 1, 

We want to sincerely thank you for reviewing our paper and submitting your kind comments about the same.

Point 1:I report only one reference used, the n 13, the authors are sure that is correct?

Response: As pointed out correctly, we have omitted the reference. 

Again, we express sincere gratitude for your appreciation. 

Regards

Nitesh Gautam

Reviewer 2 Report

This paper does a good review of the applications of ML and AI to the field of Coronary Artery Disease (CAD). This review is comprehensive regarding medical imaging for the early detection of CAD. The main focus is on standard clinical imaging modalities such as CT and MRI. They also review intravascular ultrasound (IVUS) imaging to determine the amount of atheromatous plaque built in the epicardial coronary artery. The section on risk prediction is, in particular, is an excellent review as it highlights its ranks the various variables used in ML for the prediction of heart diseases. In addition, they discuss the various issues associated with the applications and training of ML models. My only criticism is that they do not discuss other aspects of ultrasound imaging than just IVUS. For example, thoracic ultrasound imaging has been extensively used to measure ventricular capacities, detect blocked arteries, and monitor valve actions. Because of the importance of thoracic US imaging in clinical applications, I recommend that the authors add an extra section that deals with this imaging modality.

Author Response

Dear Reviewer 2, 

We want to sincerely thank you for reviewing our article and providing your valuable comments.

Point 1:Discussion about other aspects of ultrasound imaging than just IVUS.

Response: As highlighted by you, machine learning has been extensively studied in Echocardiography to assess cardiovascular diseases. A majority of articles published in the literature focus on assessing ventricular capacities, cardiomyopathies, valvular dysfunction, and assessment of regional wall motion abnormalities(RWMA). 

As suggested by you, we have included a section on echocardiography describing the recent approaches in the assessment of RWMA and its significance in CAD., which aligns with our overall topic.

While we will be delighted to write a separate review article on various advances in cardiac imaging in fields other than CAD, we feel discussing the other advances in thoracic ultrasound imaging might divert the reader from the core topic.

Once again, we express our heartfelt gratitude for your feedback! 

Regards

Nitesh Gautam

Reviewer 3 Report

The abstract has to be modified as per review paper standard.

A standard tree diagram may be included in the section 2.

The introduction section needs total revamp of ideas.

All the tables 1,2, and 3 must include the challenges as an another column. 

Discussion and conclusion  are to be separated.

Too many references./

Author Response

Dear Reviewer 3

We want to express our sincere thanks for taking the time to review our paper. As suggested by you, we have made the following changes: 

Point 1. The abstract has to be modified as per review paper standards.

Response: -The abstract has been modified according to the peer review standards. We have removed references from the abstract. 

Point 2. A standard tree diagram may be included in the section 2.

Response: -Given the extensive literature that we have reviewed(>1000 articles), it will be hard at this point to make a tree diagram. Though the tree diagram surely will increase the quality and credibility, we have made sure to include all the recent advances and significant studies in our article. 

Point 3. The introduction section needs total revamp of ideas.

Response: -Our article focussed on applications of AI in Coronary Artery Disease, we have oriented our article in a way that will be clinically relevant to the readers. Artificial Intelligence is a vast topic by itself. While we will be delighted to do another review describing the various realms of Artificial Intelligence in Medicine, we felt describing AI in more detail in the introduction would divert the reader from the core topic. For the aforementioned reasons, we have briefly described AI and have focused more on the burden of CAD and the evolution of AI. To give more clarity, we have described the outline of the article in the introduction, providing the reader with clarity about the contents of the article. 

Point 4. All tables 1,2, and 3 must include the challenges as another column. 

Response: -We have included limitations for all the studies included in tables 1,2, and 3. 

Point 5. Discussion and conclusion are to be separated.

Response: -As pointed out, we have separated the discussions and conclusions.

Point 6. Too many references

Response: -Since we have tried to summarize many studies in the review article(>120 original research articles), unfortunately, we were unable to reduce the number of references in the article. 

Once again, I sincerely thank you for your time and the quality feedback that you provided.

Regards

Nitesh Gautam

Reviewer 4 Report

Authors reviewed about the AI in Coronary Artery Disease. The authors tried to write the manuscript dividing sub-sections orderly based on diagnostic procedure starting from the risk prediction till the post treatment process.

  1. I think the authors should explain about the concept of ML & DL more in the manuscript. Tell the difference and the example how to implement. Some fundamental knowledge might be necessary for readers who are new to the AI.
  2. At the end of section 1, the authors should tell the structure of manuscript in the way that readers could know what they are going to learn from the work, and reasons.
  3. The way authors construct the sub-sections is quite difficult for readers. I think the authors should divide sub-sections by implemented AI techniques such as data analysis, image analysis or combinations of them etc. 
  4. With the approach in (3), the authors could compare the performance of AI techniques as well as their pros and cons.
  5. For me, Section 9 Discussion & Conclusion that explains about some limitations of the current techniques is not related to the previous sections.

Author Response

Dear Reviewer 4,

I would like to sincerely thank you for taking out time to review our article and your valuable feedback. Please find below a review of all the edits. 

Point 1. I think the authors should explain about the concept of ML & DL more in the manuscript. Tell the difference and the example how to implement. Some fundamental knowledge might be necessary for readers who are new to the AI.

We thank the author for the kind comments. Artificial Intelligence is a vast topic by itself. While we will be happy to do another review describing the various realms of Artificial Intelligence in Medicine, we felt describing AI in more detail in the introduction would divert the reader from the core topic. For the above-mentioned reasons, we have focused more on the burden of CAD and the evolution of AI in the introduction section. 

Point 2. At the end of section 1, the authors should tell the structure of the manuscript in the way that readers could know what they are going to learn from the work, and reasons.

Thanks for the comment. As suggested, we have described the outline of the article in the introduction, providing the reader with clarity about the contents of the article. 

Point 3 and 4. The way authors construct the sub-sections is quite difficult for readers. I think the authors should divide sub-sections by implemented AI techniques such as data analysis, image analysis or combinations of them etc. With the approach in (3), the authors could compare the performance of AI techniques as well as their pros and cons.

Thanks for the kind comments. Our article was themed around recent advances in AI in CAD. We tried to orient the article in a more clinical way, making the readers appreciate the overarching impact AI has made in recent years in all fields of CAD, ranging from genetics, risk prediction, imaging to mortality. By orienting the article around CAD, we felt it will gather more interest.

Point 5. For me, Section 9 Discussion & Conclusion that explains some limitations of the current techniques is not related to the previous sections

Thanks for the important comment. To mitigate this problem, we have included the limitations of the major studies described in the article. We have made a separate column describing the limitations of the studies in tables 1,2 and 3.

Also, we have discussed the overall limitations of AI encountered in the present, and the potential limitations that we might encounter in the future with the integration of AI and healthcare in section 9. We have tried to summarize the future challenges and the potential roadmap for AI in healthcare. 

Again, I express my sincere gratitude for your time and valuable feedback!

Regards

Subhi J Al'Aref, MD, FACC

(Corresponding author)

Round 2

Reviewer 3 Report

all the corrections are included in the paper